# Discovery of Rare Dying Radio Galaxies Using MeerKAT

Nadeem Oozeer [1,*,†] , Lawrence Rudnick [2,‡] , Michael F. Bietenholz [3,4,‡] , Tiziana Venturi [5] , Kenda Knowles [6,7] , Konstantinos Kolokythas [8] and Nceba Mhlahlo [9]

1. African Institute for Mathematical Sciences, 6 Melrose Road, Muizenberg 7945, South Africa
2. Minnesota Institute for Astrophysics, University of Minnesota, Minneapolis, MN 55455, USA; larry@umn.edu
3. Department of Physics and Astronomy, York University, 4700 Keele St., Toronto, ON M6G 3V5, Canada; mbieten@yorku.ca
4. Hartebeesthoek Radio Astronomy Observatory, SARAO, Krugersdorp 1740, South Africa
5. INAF—Istituto di Radioastronomia, Via Gobetti 101, I-40129 Bologna, Italy; tventuri@ira.inaf.it
6. Astrophysics Research Centre, University of KwaZulu-Natal, Durban 4041, South Africa; kendaknowles.astro@gmail.com
7. Centre for Radio Astronomy Techniques and Technologies, Department of Physics and Electronics, Rhodes University, P.O. Box 94, Makhanda 6140, South Africa
8. Centre for Space Research, North-West University, Potchefstroom 2520, South Africa; k.kolok@nwu.ac.za
9. Wits Centre for Astrophysics, School of Physics, University of the Witwatersrand, 1 Jan Smuts Avenue, Johannesburg 2050, South Africa; nceba.mhlahlo@wits.ac.za
* Correspondence: nadeem@sarao.ac.za
† Current address: South African Radio Astronomy Observatory, 2 Fir Street, Black River Park, Observatory, Cape Town 7925, South Africa.
‡ These authors contributed equally to this work.

**Abstract:** Dying radio galaxies represent a stage of the evolution of active galactic nuclei (AGN), during which the accreting central black hole has switched off and/or falls to such a low level that the plasma outflow can no longer be sustained. When this happens, the radio source undergoes a period of fading, the dying phase, before it disappears completely. We present the study of three potential dying radio sources using the MeerKAT radio telescope: MKT J072851.2-752743, MKT J001940.4-654722, and ACO 548B. The identification as dying radio sources came from the MeerKAT Galaxy Cluster Legacy Survey (MGCLS). We carry out a multi-wavelength analysis of the sources and derive their energetics. The ages of the sources are ∼30–70 Myr, they have magnetic fields of the order of a few μG, and they have relatively low radio power. Their potential optical counterparts are associated with massive galaxies. We show that ACO 548B, previously classified as two peripheral relic radio sources, is a dying radio galaxy. With its good sensitivity and resolution, MeerKAT is an ideal instrument to detect potential dying radio sources, and contribute to the understanding of the evolution of AGN population.

**Keywords:** radio continuum: galaxies; galaxies: nuclei; galaxies: evolution; galaxies: clusters; individual

## 1. Introduction

Extra-galactic radio sources play an essential role in areas ranging from the nature of active galactic nuclei (AGNs) and the environment of galaxies to those related to cosmology. AGNs shine over a wide range of the electromagnetic spectrum, from radio to gamma rays. In almost all of this huge energy range, AGNs are the most luminous sources in the sky. Observationally, it is well proven that some radio galaxies possess highly collimated and relativistic twin jets of matter that emerge from an AGN. The expansion of these jets produces a variety of morphologies. See [1] for a review.

Radio galaxies, primarily associated with elliptical galaxies, are supplied with energy from the AGN via plasma beams and jets, which can last over several $10^7$ years. However, after some time, the AGN activity stops and the radio source fades, the dying phase, before it disappears completely [2]. In the dying phase, the radio core, jets and hotspots

will disappear. The fossil radio lobes, however, remain detectable despite the radiative losses of the relativistic electrons. We expect the fading lobes to have a very steep spectral index[1] ($\alpha > 1.3$) and a convex radio spectrum, characteristic of a population of electrons which have radiated away much of their original energy [3]. These sources will also show low surface-brightness extended emission. Giovannini et al. [4] suggested that, many dying radio sources may not show the presence of a core due to lack of adequate resolution and sensitivity. However, we believe the core may be visible given the sensitivity and resolution of MeerKAT.

Brienza et al. [5] found that the fraction of remnant/dying sources is <6–8% of the entire radio source population, while Hurley-Walker et al. [6] estimated the space density of the low brightness (dying) phase of radio galaxy evolution as $7 \times 10^{-7}$ Mpc$^{-3}$ from their Murchison Widefield Array (MWA) observations. The estimates of from the MWA are roughly seven times greater than that of the estimates made from the Sydney University Molonglo Sky Survey (SUMSS) [7]. One the reason may the high surface brightness sensitivity of the MWA compared to that of SUMSS.

Dying radio galaxies represent an intriguing class of sources that are still largely unexplored. Having a bigger sample of these hard-to-find sources will allow us to understand the various stages of the AGN life cycle. There is also a possibility that, after some time, the AGN gets re-triggered and starts emitting radio plasma along a new jet axis. This new activity episode (see [8] for a review) could have been triggered soon after the axis change or long after the axis change. Saripalli & Roberts [9] used this as one of the explanations for the S-, X- and Z-shaped radio source, but Cotton et al. [10] ruled out models that invoke jet re-orientation or two independent jets for explaining the X-shaped radio source PKS 2014-55. However, even if all AGN eventually get re-triggered, some AGN will be in the dying stage. We therefore expect, some fraction of AGN to be in the dying phase at any given time.

There is another type of radio source that has characteristics similar to those of dying radio sources. They are known as peripheral radio relics. Radio relics are diffuse extended sources that are detected at the periphery of cluster mergers. In the radio regime, they are characterised by irregular, at times elongated morphologies, steep spectral indices ($\alpha > 1$), and high linear polarization ($\sim$20–30%). Elongated relics do not show any evident substructures, and in some case their transverse size is very small. Radio relics trace shock waves occurring in the intracluster medium (ICM) during cluster merger events, far from the cluster center. Diffuse extended radio sources with a more regular and roundish structure have been detected off-center in clusters [11]. Such sources, that can be confused with dying radio sources, have been found close to AGN but however, they are always located only on one side of the First Ranked Galaxy (FRG) [12]. The main distinction between radio relics and dying radio sources is that radio relics originate from the ICM, while dying radio sources are linked to the AGN. Therefore, a radio relic should not have a radio core or hotspots nor an optical host.

Radio phoenix sources are another class of radio sources that are related to AGN, found in merging clusters, and were previously confused with relics. As mentioned by Kempner et al. [13], these sources begin their lives as normal radio galaxies, but with a population of electrons that have aged so much that they no longer emit synchrotron radiation. When shocks from a cluster merger pass through the old AGN lobe, compression of the fossil radio plasma can re-energize the electrons, so they shine again at radio wavelengths. The main observational property that the sources have in common is the AGN origin of the plasma and their ultra-steep radio spectra. Often these phoenices display irregular filamentary morphologies. They have relatively small sizes of at most several hundreds of kpc (see [14] for a review).

In this paper, we shall use the definition of dying radio sources as given by Murgia et al. [15]. We classify a radio source as dying if either the fossil lobes are detached from the AGN and there is no evidence of nuclear activity, or if some kind of nuclear activity is present but the fossil lobes dominate the total source's radio luminosity.

A combination of sensitivity and resolution is required to detect these dying radio sources. MeerKAT [16] has proven to be an ideal telescope to probe such types of radio sources with its very good sensitivity, resolution and short baselines to pick up diffuse emission. This paper will focus on three targets; MKT J072851.2-752743, MKT J001940.4-654722, and ACO 548B that were observed using MeerKAT. The three targets were observed as part of the MeerKAT Galaxy Cluster Legacy Survey (MGCLS), described by Knowles et al. [17], which observed a heterogenous sample of galaxy clusters with no mass or redshift selection criteria.

The goal of this paper is to dissect these new sources and address various physical and environmental parameters. We organise this paper as follows. In Section 2 we summarise the observation and data processing carried out to produce the images, and in the Section 3, we show the multi-wavelength analysis, followed by a discussion from our findings in Section 4 and provide a conclusion.

Through out this paper, we shall use a flat vacuum dominated universe with $H_0 = 67.8$ km s$^{-1}$ Mpc$^{-1}$, $\Omega_m = 0.308$, $\Omega_{vac} = 0.692$ [18].

## 2. Observations and Data Reduction

The MGCLS observations were in the frequency band 856–1712 MHz, using MeerKAT. The MeerKAT array consists of 64 antennas, each 13.5 m in diameter with a dense core with 48 antennas located within a diameter of 1 km, and the remaining 16 antennas spread out to give a maximum baseline length of 8 km. Our observations had a bandwidth of 856 MHz centred at 1284 MHz, split into 4096 channels. The data processing and reduction are explained in depth in Knowles et al. [17] and references therein.

The final products from this survey (MGCLS DR1 data products) consist of primary products and enhanced products. The primary products consist of images, with one image at the center frequency of 1284 MHz, one spectral image calculated between 908 and 1656 MHz, and 14 sub-band images. The central frequencies for the sub-band images are at 908, 952, 996, 1044, 1093, 1145, 1200, 1258, 1318, 1382, 1448, 1482, 1594, and 1656 MHz. Two sub-band images (1200 and 1258 MHz) are not available as they are blanked due to radio frequency interference (RFI). These primary product images are not primary beam corrected and covered $\approx 3°$. The enhanced images, on the other hand, are primary beam corrected. Full stokes (I, Q, U, and V) with full resolution (7.5–8″ at Full Width Half Maximum (FWHM) and convolved (15″ FWHM) products are produced. For polarization images, the calibration including removal of leakage and polarization angle determination was as described Knowles et al. [17]. Imaging in Stokes Q and U used the same frequencies as described above. Low brightness, large scale ripples were present in some frequency channels, and there are likely some residual problems with off-axis leakage terms at the highest frequencies, so the results presented here should be viewed as preliminary.

## 3. Analysis and Findings

We inspected 115 primary data product images, to search for non-cluster related diffuse emission. During this search, we found some relatively weak diffuse emission in three of the images. To increase the detectability of the lowest brightness emission, we used the multi-resolution filter of Rudnick [19] which enables very low brightness emission to be seen by first removing brighter, smaller-scaled features. We applied this filter to the 1.284 GHz enhanced images, using a window size of 19 pixels (23.75″); this removes ∼50% of the flux density of structures with a size of ∼25″ in either dimension. Smaller scale structures have correspondingly higher fraction of the flux density removed. After filtering, we convolved the images with a 25″ FWHM circular Gaussian. We used the filtered images only to highlight the diffuse emission in our figures, and they are not used in any analysis.

Furthermore, since we are interested in characterising the diffuse emission, the spectral indices and other parameters were calculated using the primary beam corrected, 15″ FWHM, enhanced image cubes, unless stated otherwise. We describe the results in the following sections.

### 3.1. MKT J072851.2-752743

This source was spotted in the observations of the cluster PSZ1 G287.05-23.21, which is at a redshift, $z$, of 0.111. The full resolution MeerKAT radio image for the source, Figure 1, shows two separate diffuse blobs that we labelled Blob 1 (centred on J2000 RA $07^h$ $28^m$ $51^s.2$, Dec $-75°$ $27'$ $43''.28$) and Blob 2 (centred on J2000 RA $07^h$ $28^m$ $20^s.37$, Dec $-75°$ $26'$ $11''.03$). A bright source, which is shown in the white box in Figure 1 is located at an RA $07^h$ $28^m$ $32^s.43$, and Dec $-75°$ $27'$ $40''.03$. The flux density before primary beam correction is 530 µJy, which is therefore a lower limit on the true flux density.

The bright source is $\sim46''$ from the midpoint between Blob 1 and Blob 2, and is the only radio source which might be the core nearby. Using the MGCLS source catalog [17], we can calculate that the chance of finding a unrelated source of $\geq 530$ µJy in a region of radius $46''$ on this image is $\sim4\%$ (this is a conservative estimate, since the 530 µJy is before primary-beam correction, which correction would decrease the odds of a chance coincidence). We therefore assume that the bright source is in fact the core and physically related to Blob 1 and Blob 2 in what follows.

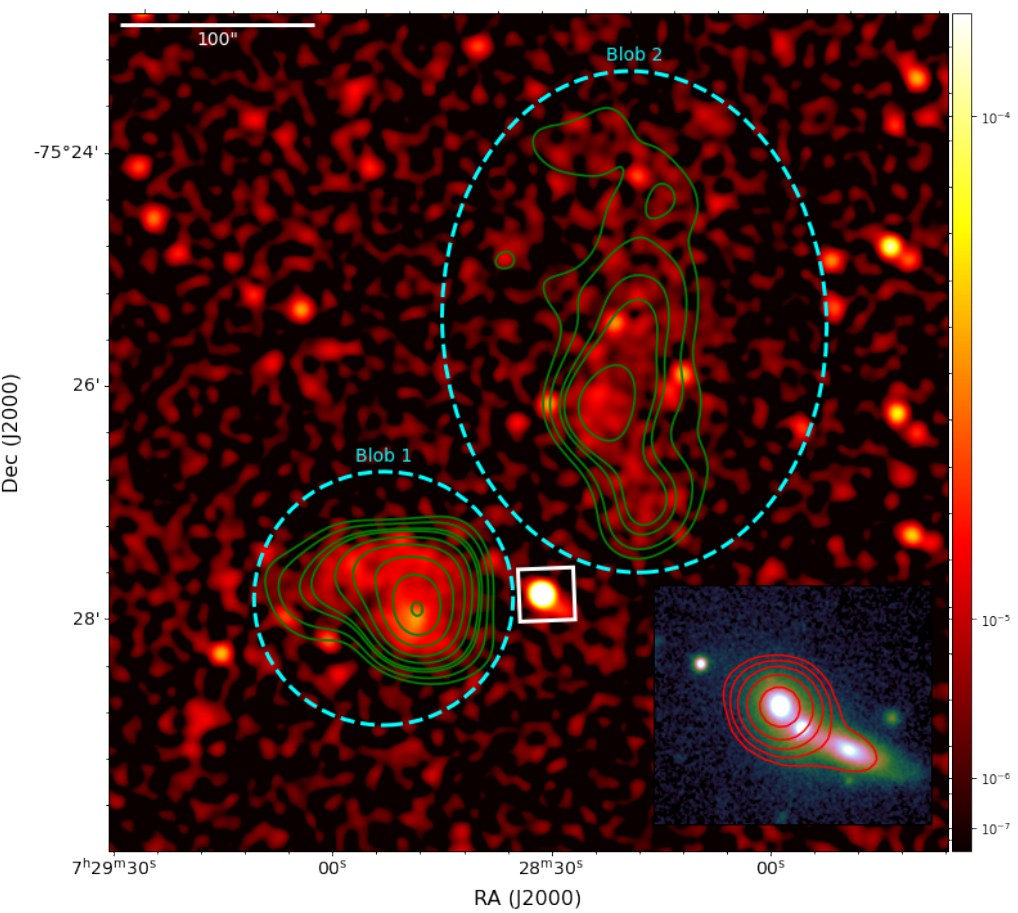

**Figure 1.** The colour image is the non-primary beam corrected full resolution MerKAT image of MKT J072851.2-752743, at a resolution of $\sim8''$. The colour bar is in units of Jy beam$^{-1}$. The green contours show the filtered image, and are at [22, 30, 42, 48, 64, 84, 114, 144, 169] µ/($25''$ beam). Two regions of low surface brightness diffuse emission are marked Blob 1 and Blob 2, and the strongest source in the field is the one in between these blobs, marked by the white box. The inset image shows the MeerKAT full resolution image (FWHM $7.5'' \times 7.4''$, at position angle, p.a., 36°) as contours on the Dark Energy Survey (DES) optical i-band image. The contours are at [2.5, 5, 10, 20, 40] $\times$ 7.5 µJy beam$^{-1}$.

To better determine the properties of the faint, diffuse emission, we used the lower resolution ($15'' \times 15''$) enhanced-product MGCLS images. We draw a region along the $3\sigma$ contours, where $\sigma$ is the local root mean square (rms) of the noise, around the source in Blob

1. We then extracted the flux density in this region. We used the same region through all the sub-band enhanced images and then calculated the fitted spectral index to be $2.6 \pm 0.3$. This relatively steep spectral index for the diffuse emission is consistent with a fading lobe. The spectral index for the core and Blob 2, on the other hand, could not be calculated since they are outside the available primary-beam corrected image.

### 3.1.1. Optical Identification

Using the position of the potential core, shown by a white box in Figure 1, we searched for optical and infrared (IR) counterparts. We found that the potential core seems to be associated with an IR source, WISEA J072832.45-752740.0 at a $z = 0.0138$ [20], and we therefore take the lobes to also be at that redshift.

The potential radio core is slightly resolved in the full resolution image, with an angular size (after convolution with the CLEAN-beam) of $7.9'' \times 7.3''$ at a position angle, p.a., of $51°$. It has a total extent of $\sim 4.9$ kpc with a morphology of a wide-angle tail as shown in Figure 1. Furthermore, the WISE image shows extended emission from likely interacting galaxies in a common envelope. Evidence of complex core structure is also found in the possible dying radio galaxy J1828 + 49, which Brienza et al. [21] suggest may be responsible for shutting off the AGN jets. Since the potential core is outside the primary beam corrected image, we can not give a reliable value of the flux density.

Using the W1 band magnitude, we estimated the mass of the WISE galaxy, using the method explained in Wen et al. [22], to be $1.539^{+0.037}_{-0.036} \times 10^9 M_\odot$. The Wise W1$-$W2 colour, which primarily samples flux density from stellar photospheres, was found to be 0.139 mag. This low colour difference suggests a system of primarily old stars. The longer WISE wavelengths (W3 and W4) are more sensitive to warm dust emission heated by stars or of the dusty torii surrounding some accreting black holes [23]. The W3$-$W4 colour is 2.05 mag that puts this galaxy in the spiral type with a potential remnant/re-started category, where this is a phase when the activity within the galaxy stops or substantially decreases.

### 3.1.2. Source Energetics

Since Blob 1 is within the primary-beam corrected image, we could calculate its physical parameters. The maximum angular diameter was $128''$, which, at $z = 0.0138$, implies a linear size of 37 kpc. From the various parameters, and a flux density of $2.94 \pm 0.63$ mJy, we calculated the spectral luminosity of the diffuse emission at 1.4 GHz to be $L_{1.4\mathrm{GHz}} = 1.07 \times 10^{28}$ erg s$^{-1}$ Hz$^{-1}$. We calculated the volume of the source assuming a prolate ellipsoidal geometry with the major and minor axis projected linear size. Using the equations from [24], we computed the total energy density of the synchrotron-emitting component to be $5.3 \times 10^{-12}$ erg cm$^{-3}$, and the magnetic field to be $\sim 7.5\,\mu$G, with the assumption that the plasma is in an equipartition condition between particles and the magnetic field. The derived magnetic field is quite sensitive to the spectral shape as mentioned in Brienza et al. [21]. We calculated the radiative lifetimes against a combination of synchrotron and inverse Compton cooling, using the equation in Parma et al. [25], to be $\sim 30$ Myr. Since we are in a steep spectral energy regime, we use a break frequency to be 1.7 GHz in all of our calculations.

### 3.2. MKT J001940.4-654722

MKT J001940.4-654722 is located in the MeerKAT field of Abell 2746, which is at $z = 0.1545$. MKT J001940.4-654722 consists of two faint low surface brightness diffuse radio sources that we again label as Blob 1 (centred on J2000 RA $00^\mathrm{h}\,20^\mathrm{m}\,01^\mathrm{s}.93$ Dec $-65°\,48'\,04''.08$) and Blob 2 (centred on J2000 RA $00^\mathrm{h}\,19^\mathrm{m}\,10^\mathrm{s}.59$, Dec $-65°\,46'\,55''.19$). Figure 2 shows the MeerKAT low-resolution image as contours and the optical image. The overall size of the source (Blob 1 and Blob 2 included) as measured end-to-end from the second contour level is around $450''$.

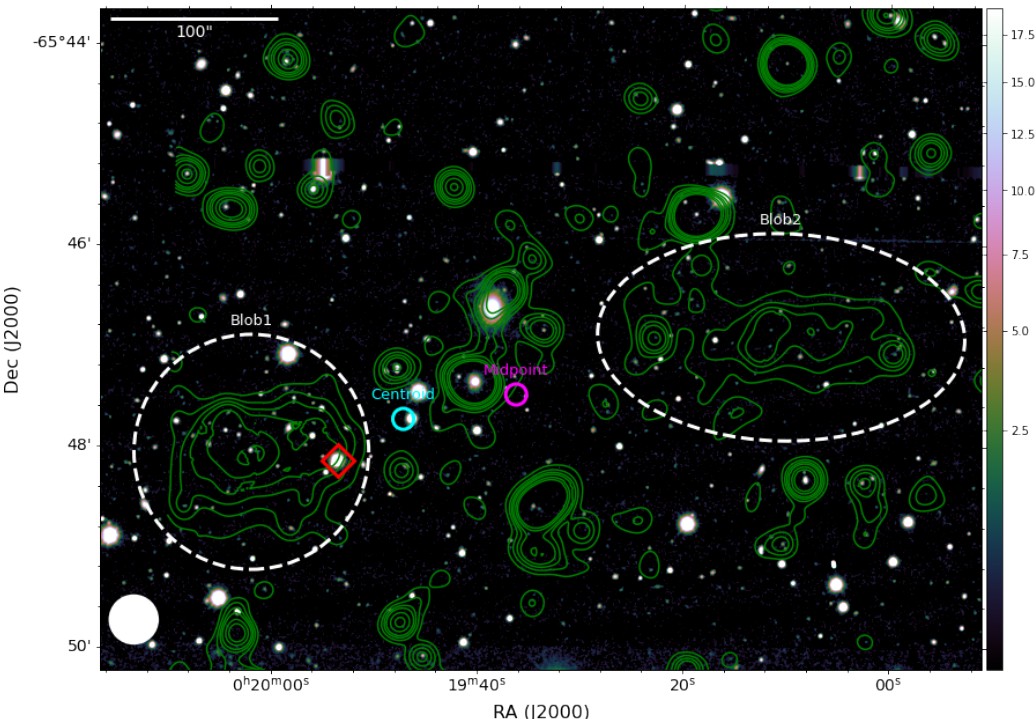

**Figure 2.** The green contours show the MeerKAT primary beam corrected image of MKT J001940.4-654722 at a resolution of 15″ FWHM. The resolution is shown by the filled white circle in the bottom left, and the contours levels are at $[2.5, 5, 7.5, 10, 15, 20] \times 10$ μJy beam$^{-1}$. The colour scale shows the false colour g, r, i-composite image from the DES optical survey, in units of electron counts. The radio centroid, and midpoint are also shown. The red diamond indicates the presence of a stellar object which is not associated with the diffuse radio emission.

Since we are more interested in the diffuse emission, we used the low resolution (15″ × 15″) primary beam corrected images to extract the radio properties of the two blobs. Using the full resolution as contours onto the low resolution image, we draw regions around the diffuse blobs to extract the flux. This allows us to exclude the point sources in the parameter extraction. The extracted parameter values are tabulated in Table 1. The two diffuse Blobs have similar steep spectral indices.

**Table 1.** Parameters of the diffuse radio emission, determined from the low resolution primary beam corrected image of MKT J001940.4-654722. The columns are as follows: Col. 1—Region name; Cols. 2, 3—Right Ascension and Declination of the region or source; and cols. 4, 5—Maximum and minimum size of the source as measured end-to-end from the second contour level (for the core this is the FWHM of the major and minor axes); Col. 6—Flux density at 1.284 GHz, and Col. 7—the spectral index calculated between 0.908 GHz and 1.565 GHz.

|  | RA (J2000) h:m:s.s | Dec (J2000) d:m:s.s | Bmax ″ | Bmin ″ | $S_{1.284\ GHz}$ mJy | $\alpha_{0.908\ GHz}^{1.656\ GHz}$ |
|---|---|---|---|---|---|---|
| Blob 1 | 00:20:01.93 | −65:48:04.08 | 131 | 90 | $3.16 \pm 0.47$ | $2.5 \pm 0.2$ |
| Blob 2 | 00:19:10.59 | −65:46:55.19 | 221 | 78 | $2.34 \pm 0.36$ | $3.2 \pm 0.4$ |
| Core | 00:19:40.37 | −65:47:21.96 | 16 | 15 | $1.44 \pm 0.32$ | $0.8 \pm 0.1$ |

### 3.2.1. Optical Identification

Figure 3 shows the filtered MeerKAT image of MKT J001940.4-654722 as contours, that show the extent of the diffuse emission. There are a number of faint optical sources ranging from red optical (R) magnitude 19 to 21 within this field. In the region of Blob 1, we detected a relatively bright optical source. This source is associated with a stellar object, WISE J001953.60-654809.2. Blob 2 contains no clear optical/IR sources within the

diffuse radio emission, except for the one associated with the unresolved radio source at the western edge of Blob 2.

We calculated the mid-point, and the radio centroid using the flux densities of Blob 1 and Blob 2. The mid-point is the point midway between peak-brightnss positions of the two blobs. The locations of the midpoint and centroid are shown in Figure 3. We used the midpoint and centroid coordinates to search for potential optical counterparts. The source closest to the radio centroid is WISEA J001946.36-654744.2, but it shows a stellar nature, and we found no sources around the radio mid-point.

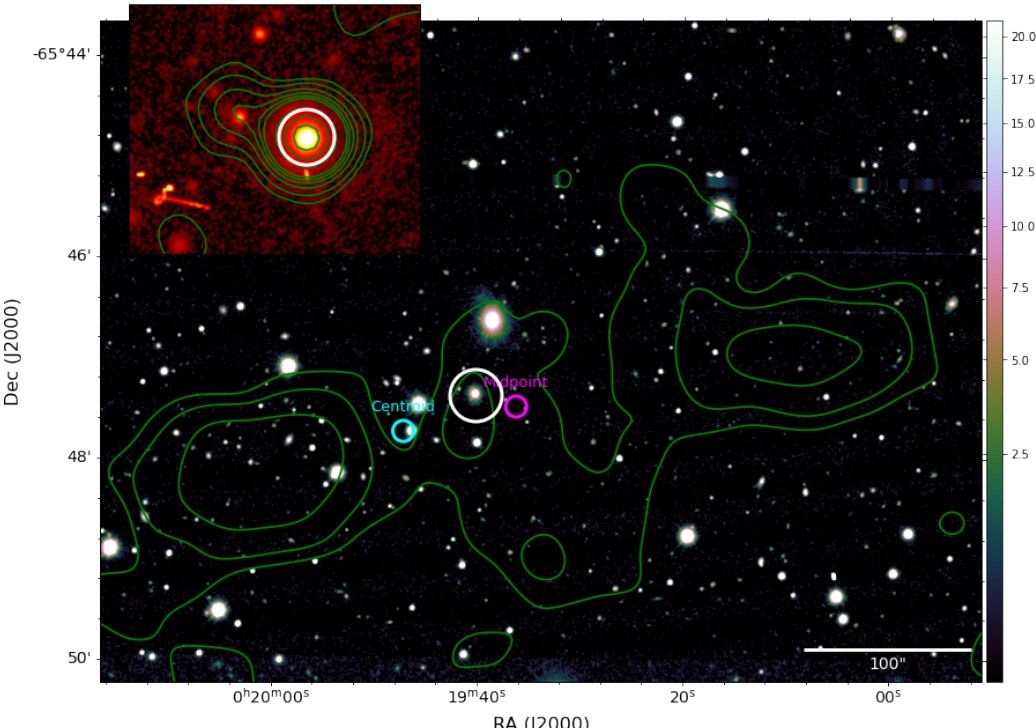

**Figure 3.** The contours show the primary-beam corrected filtered image of low brightness emission in MKT J001940.4-654722 at [1.2, 3.1, 7.3] $\times$ 10 μJy/($25''$ beam) after removal of emission on smaller scales. Much of the structure visible in Figure 2 has been subtracted out. Note that a faint bridge connecting the lobes can now be seen. The image is a false colour g, r, i-composite image from the DES optical survey. The units for the colour scale are electron counts. The white circle shows the location of the potential core for this system. The inset at upper left shows an expanded view of the potential core. The inset radio contours are at [2.5, 5, 10,15, 20, 25, 50, 75, 100] $\times$ 10 μJy beam$^{-1}$.

The white circle on Figure 3 shows a compact radio source, which is close to the midpoint, and we will assume this radio source is associated with the infrared source WISE J001940.23-654722.1. The zoomed region around the potential core, inset image in Figure 3, shows that the core also has more than one radio component, perhaps associated with a companion galaxy.

To determine the properties of the host galaxy, we investigated the infrared colours and the WISE colour–colour magnitude of the potential core. The Wise W1−W2 magnitude has a low value of 0.274 mag, unlikely for an AGN, while the W3−W4 band magnitude is around 3.447 mag, as obtained using the values from Cutri et al. [26]. On the WISE colour–colour diagram, this source is situated at the edge of a spiral and starburst group (see [23,27]). We calculated the photometric redshift of the source to be 0.22 using the Dark Energy Camera Legacy Survey Data Release 9 (DECaLS DR9) photometry. The package zCLUSTER[2] was used to estimate the photometric redshift using multi-band optical and infrared photometry. zCLUSTER does template fitting as explained in [28]

and [29] . Assuming the diffuse emission to be at the photometric redshift, and using the angular size as measured from the filtered image, the linear size of the source is ∼880 kpc.

### 3.2.2. Source Energetics

Using the photometric redshift of $z = 0.22$, and the values from Table 1 , we calculated the energetic parameters for both Blob 1 and Blob 2 using a method similar to that explained in Section 3.1.2. The brightness temperature of Blob 1 is ∼205 mK and the radio spectral luminosity at 1.4 GHz is $L_{1.4GHz}= 3.8 \times 10^{30}$ erg s$^{-1}$ Hz$^{-1}$. Using a cylindrical volume of the source, the total energy density of Blob 1 is found to be $1.5 \times 10^{-12}$ erg cm$^{-3}$, and the magnetic field to be ∼4.0 μG. The synchrotron lifetime is ∼48 Myr.

For Blob 2, which has a brightness temperature of ∼50 mK, the radio spectral luminosity at 1.4 GHz is $L_{1.4GHz} = 2.7 \times 10^{30}$ erg s$^{-1}$ Hz$^{-1}$. The total energy density is found to be $23.0 \times 10^{-12}$ erg cm$^{-3}$, and the magnetic field is ∼15.6 μG. The synchrotron lifetime is ∼51 Myr.

The core has brightness temperature of ∼7.1 K, and a radio spectral luminosity at 1.4 GHz of $L_{1.4GHz} = 2.0 \times 10^{30}$ erg s$^{-1}$ Hz$^{-1}$. Using the W1 magnitude of the core, we calculated the galaxy mass to be $137.5 \pm 4 \times 10^{9} M_{\odot}$, which is consistent with the stellar masses of massive starburst galaxies [30].

### 3.3. ACO 548B

Finally, we provide a new look at Abell 548B (ACO 548B), a well known galaxy cluster at a $z = 0.0415$. It was analysed by Feretti et al. [31], who classified it as a double relic. The MeerKAT image suggests that we may be dealing with a large dying radio galaxy ∼850″ in extent (linear size of 650 kpc at $z = 0.0356$). Figure 4 shows the diffuse emission from this source at a resolution of 7.5″. We identify two components in the diffuse emission. The North-East (NE) lobe is centred on J2000 RA 05$^h$ 45$^m$ 22$^s$.10, Dec −25° 47′ 29″.88, while the South-West (SW) lobe is centred on J2000 RA 05$^h$ 44$^m$ 05$^s$.00, Dec −25° 50′ 31″.40. The brightness temperature of the NE lobe is ∼243 mK, and that of SW lobe is ∼575 mK.

There is a radio galaxy between the two lobes (marked with a cyan circle in Figure 4), which is associated with the 6dFGS source g0545049-254740, at $z = 0.0356$ [32]. Knowles et al. [17] showed that this source has a double morphology from the recent Very Large Array Sky Survey (VLASS) [33] 3 GHz map at 2.5″ resolution. Another compact radio source is within the NE lobe, and is associated with the spiral galaxy 6dFGS g0545221-254730 at $z = 0.038$. This latter compact source has a a wide-angle-tail (WAT) morphology, swept towards the West by ∼80 kpc, see inset image in Figure 4. It is not clear whether there is a physical connection to the dying radio galaxy [17]. There is another spiral galaxy at the western extremity, which is likely unrelated to the diffuse radio emission, and is associated with the 6dFGS g0544374-255335, at $z = 0.039$.

Finally, to the South, nearer the cluster centre, is a 220 kpc long narrow-angle-tail (NAT), shown in the bottom of Figure 4, associated with 6DFGS g0545275-255510, at $z = 0.042$.

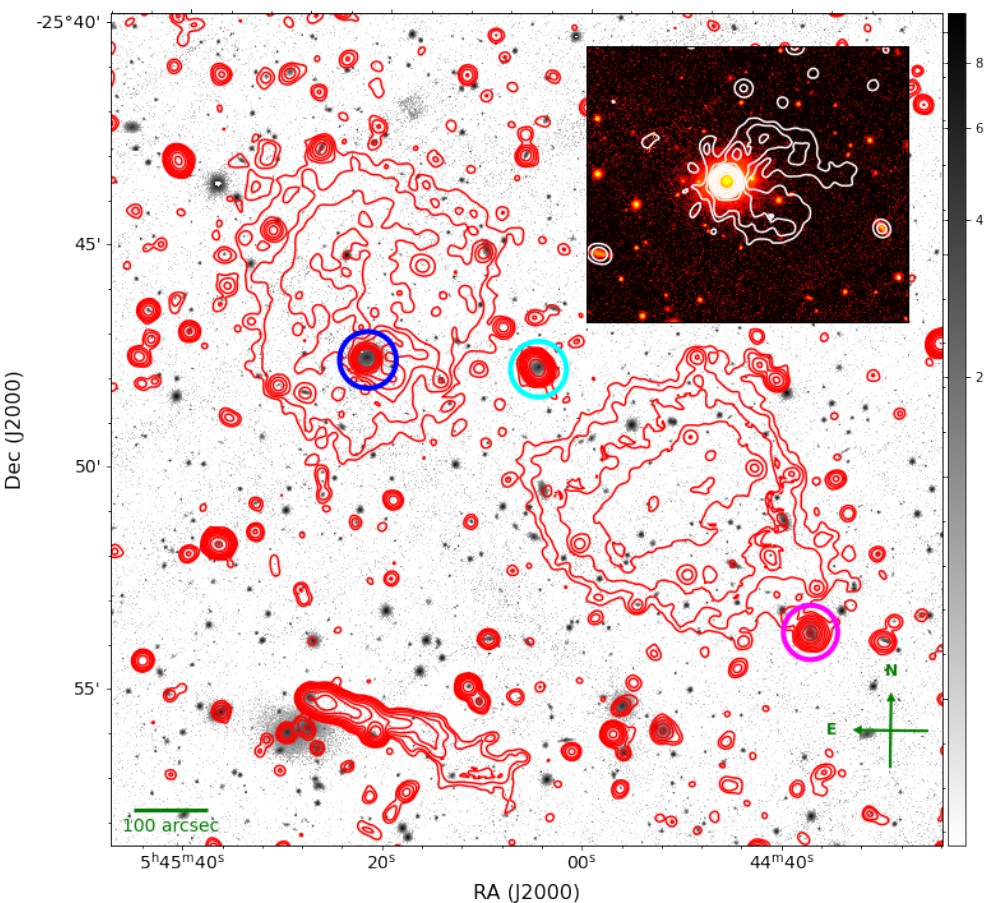

**Figure 4.** Radio contours of MeerKAT low resolution (15″ FWHM) primary beam corrected image of ACO 548B with PanSTARRS DR1 i-band greyscale. The contours are the 1.284 GHz radio emission and are at [3.5, 7, 14, 28, 56, 102] × 10 μJy beam$^{-1}$. The NE lobe has a WAT source associated with an 9.7 Rmag [32] optical galaxy MCG −04.14.021 (blue circle) at a $z = 0.038$. This source is associated with the Abell cluster 548B. The SW lobe has a point source (magenta) at the periphery which is associated with the 6dFGS source g0544374-255335 at $z = 0.039$. The cyan circle, in between the two lobes indicates a radio source that is associated with the 6dFGS source g0545049-254740, which is at $z = 0.0356$. A narrow-angle tail (NAT) source is found to the South, nearer to the cluster center. The upper right inset shows the radio image of the WAT source as contours onto the same optical image. In this inset, we smoothed the MeerKAT image to a 25″ FWHM. The contours are at [−1, 1, 2, 4, 8, 16, 32, 64, 128, 256] × 75 μJy beam$^{-1}$.

We used a spectral index image which was a product from the OBIT[3] pipeline that produced the MeerKAT images, which we show in Figure 5. The pipeline fits the spectra pixel by pixel, with a default spectral index of 0.6 used if a satisfactory fit is not obtained. Spectra beyond ∼36′ from the field centre are generally not reliable because pointing errors change the primary beam correction [17]. We found the values in this case reliable since out target was within the 36′ of the field centre. We extracted the spectral index distribution from the MeerKAT band by drawing regions along the 3 × σ contours from the 15″ resolution image, but excluding the compact sources within the bubble.

The diffuse lobes clearly have very steep spectral indices (Figure 5) . The average spectral index from the pipeline product for the NE lobe from the spectral index image is around 2.2, and that for the SW lobe is 1.7. The spectral index we obtain for the NE and SW components using least square fitting are 2.4 ± 0.1 and 2.2 ± 0.1 respectively. These values are consistent with the results obtained from the VLA L-band observations in [31]. The central core has a flat spectral index (0.5 ± 0.1 between 952 and 1656 MHz), which is consistent with lower frequency spectral index of $\alpha^{231\text{ MHz}}_{72\text{ MHz}} = 0.5 \pm 0.1$ calculated by

Hurley-Walker et al. [34]. The MeerKAT spectral index for the WAT core within the NE lobe is $0.4 \pm 0.1$, while [34] calculated the spectral index to be $\alpha_{72 \text{ MHz}}^{231 \text{ MHz}} = 0.9 \pm 0.1$.

The steep spectral indices of the diffuse emission indicate potential lobes from a dying radio source. Owing to the complexity of the diffuse emission, and the embedded unresolved sources, it is not easy to extract energetics from the current images. One would need to re-image the visibility in-depth to remove these sources and make new images of the diffuse emission. However, we provide an upper estimate of the energetics of the two diffuse lobes, using $z = 0.0356$ and a spherical size as measured from the filtered image. The NE lobe has a flux density of around 36 mJy, while the SW lobe has a flux density of 75 mJy. The NE lobe has a radio spectral luminosity $L_{1.4\text{GHz}} = 9.0 \times 10^{29}$ erg s$^{-1}$ Hz$^{-1}$. The total energy density, assuming a spherical volume, is $1.0 \times 10^{-12}$ erg cm$^{-3}$, and the magnetic field is $\sim$3.3 μG. The SW lobe has a radio spectral luminosity $L_{1.4\text{GHz}} = 1.9 \times 10^{30}$ erg s$^{-1}$ Hz$^{-1}$. The total energy density is $0.9 \times 10^{-12}$ erg cm$^{-3}$, and the magnetic field is $\sim$3.1 μG. The radiative lifetime of the both lobes is $\sim$67 Myr.

The brightness temperature of the potential core source is $\sim$113 K. The radio spectral luminosity $L_{1.4\text{GHz}} = 1.02 \times 10^{30}$ erg s$^{-1}$ Hz$^{-1}$. The mass of the optical counterpart, which is associated with a S0-a galaxy, is $38 \pm 1 \times 10^9 M_\odot$.

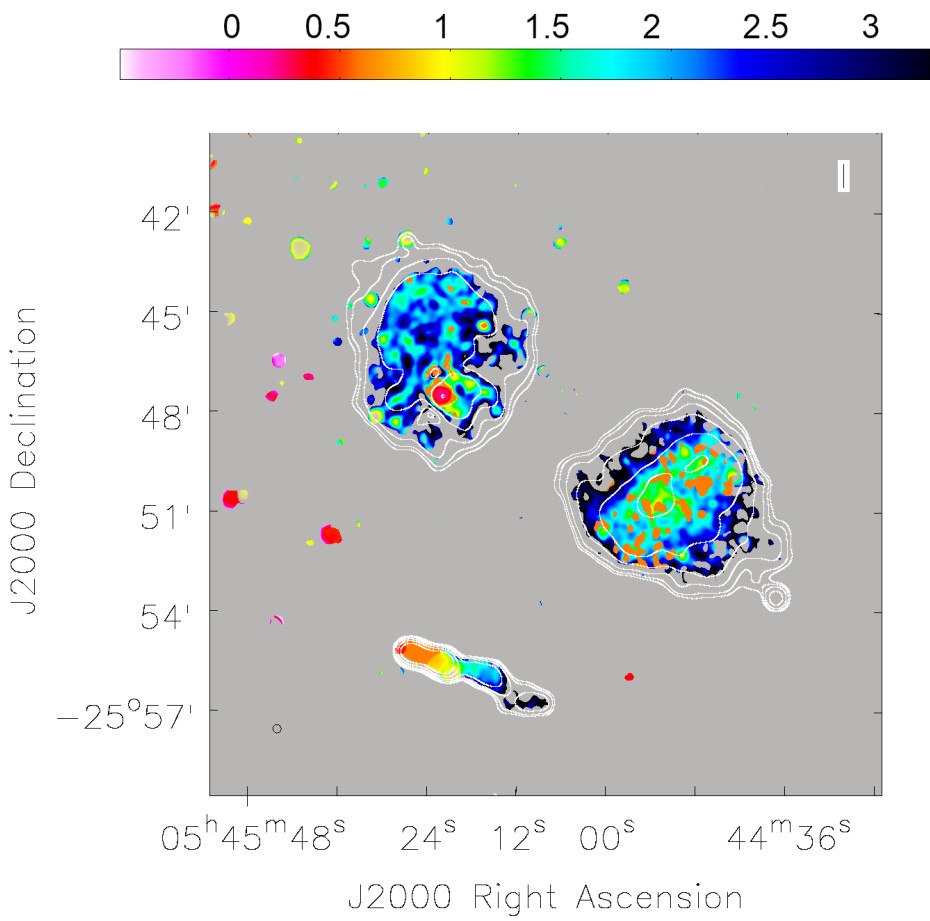

**Figure 5.** The colourscale shows the spectral index of ACO 548B as obtained from the pipeline. The contours are at $[3.5, 7, 14, 28, 56, 102] \times 10$ μJy beam$^{-1}$ and show the radio continuum brightness of the filtered enhanced MeerKAT image.

We also made polarization images of ACO 548B, and the results are shown in Figure 6, made by combining the Q and U maps in three bands, at 966, 1044 and 1093 MHz. The results are preliminary because individual sub-band Q and U maps had low level large-scale

ripples, and careful removal of these is beyond the scope of the current paper. This initial look reveals that around frequencies of ~1 GHz, the average fractional polarization of the west lobe and the brighter portions of the east lobe are ~13%, reaching up to 40% at locations on the leading edges. Using three different pairs of sub-bands we found a consistent value for the average rotation measure of 23–25 rad m$^{-2}$ for each lobe; because of the limited accuracy of rotation measures in each individual beam, we corrected the angles using an average of 23.5 rad m$^{-2}$. The Galactic foreground RM in the direction of Abell 548 is 25 ± 8 rad m$^{-2}$ [35], so we find no evidence for a net Faraday contribution from the cluster. The preliminary de-rotated magnetic field directions in the lobes show a generally circumferential orientation around the leading edges and a large region of organized field orthogonal to the major axis in the interior of the western lobe. These circumferential fields are very similar to those in the circular lobes of the FRI sources studied by Laing et al. [36]; since those sources have jets, it raises the issue of whether the effects of ACO 548B's earlier jets are still reflected in the lobe dynamics.

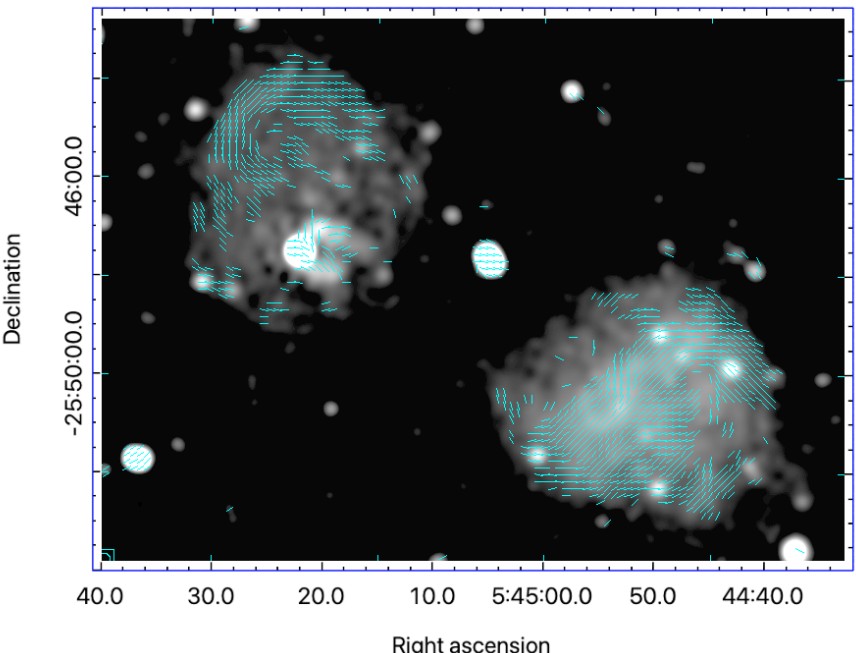

**Figure 6.** The polarization of ACO 540B. Vectors indicating the direction of the magnetic field after correction for an average rotation measure of 23.5 rad m$^{-2}$, overlaying a greyscale total intensity image. Vectors are shown only when the bias-corrected polarized signal:noise was above ~10.

### 3.4. X-ray View

From their relatively low resolution (15″ × 30″) and sensitivity radio observations, Feretti et al. [31] suggested that the radio galaxy lobes in the north of the NAT and the diffuse emission to the south were merger-related relic structures outside of the bright X-ray region of the cluster. We show a smoothed (5 Gaussian) image, Figure 7, of the XMM-Newton data that Solovyeva et al. [37] used together with our new MeerKAT primary-beam corrected 15″ resolution image. We found the NAT to be around the X-ray cluster centre. The X-ray emission centroid is around 178 kpc and 453 kpc from the NE and SW diffuse lobes radio centroids, respectively.

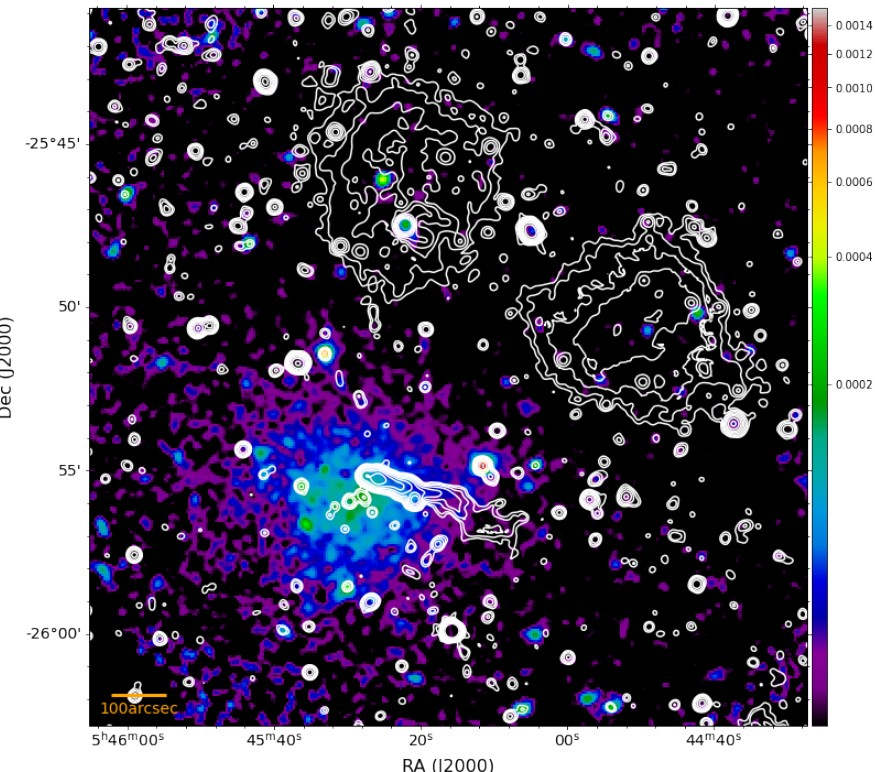

**Figure 7.** The low resolution (15″) MeerKAT primary beam corrected image of ACO 548B as contours, overlayed onto a smoothed XMM-Newton greyscale image. The contours are at [3.5, 7, 14, 28, 56, 102, ...] × 10 μJy beam$^{-1}$.

From their X-ray analysis, Solovyeva et al. [37] showed that ACO 548B is a cluster in a complex dynamical state. From their data, it was impossible to establish if the central relic that they referred to as C, the NAT in our image, is associated with any shock or X-ray feature. The MeerKAT radio image shows that the "supposed relic C" is a NAT ramming into the hot cluster plasma. The NAT is clearly associated with the optical galaxy 6DFGS g0545275-255510, and we can therefore confidently rule out the possibility of it being a relic.

Furthermore, Feretti et al. [31] mentioned that in the radio regime, the diffuse lobes (NE and SW) show around 30% linear polarisation, and the NAT showed around 7% linear polarisation. Relics are known to be highly polarized, see Feretti et al. [12], and van Weeren et al. [14] for a review.

Nakazawa et al. [38], using deep Suzaku X-ray observations, measured the Intra Cluster Medium (ICM) properties to a radius 16′ from the cluster centre, well beyond the two relic candidates. They found that the ICM morphology, temperature, and entropy do not show strong structure, with marginal evidence for slight temperature variation. They also conclude that it is natural that there is some cooler gas in a region within the relics rather than assuming hotter gas. By analysing the SW lobe, they put a lower-limit on the magnetic field strength of > 0.5 μG, which is consistent with the equipartition field of 0.9 μG measured by Solovyeva et al. [37]. Our upper limit for the magnetic field is 3.5 μG.

## 4. Discussion

We have analysed three potentially dying radio sources. We summarise the findings from the energetics analysis in Table 2. All three dying radio sources show diffuse emission with very steep/ultra-steep spectral indices, which indicate an old population of electrons.

Although relics also have steep spectral indices, the relatively circular appearance of the sources described here is much more consistent with old radio lobes than the more elongated structures typically associated with relics.

Furthermore, the radio spectral luminosity of the three sources is similar to that of low power Fanaroff-Riley Class I (FRI) [39] radio sources, with the exception of MKT J072851.2-752743. This source has a very low radio spectral luminosity, typical of a regular radio galaxy.

**Table 2.** Table summarising the energetics of all the 3 dying sources discussed in this paper. Note: †—When the target was outside the primary beam corrected image, we did not extract the energetics of the source. The spectral indices were calculated by fitting a power law to the flux density, which was extracted by drawing regions around the source of interest. * The magnetic field is calculated with the minimum energy assumption.

| Source Name | RA [h m s.s] | Dec [d m s.s] | $\alpha^{1656\ \text{MHz}}_{952\ \text{MHz}}$ | $L_{1.4\text{GHz}} \times 10^{30}$ [erg s$^{-1}$ Hz$^{-1}$] | $E_{\text{tot}} \times 10^{-12}$ [erg cm$^{-3}$] | B * [μG] | $t_{\text{rad}}$ [Myr] |
|---|---|---|---|---|---|---|---|
| **MKT J072851.2-752743** | | | | | | | |
| Blob 1 | 07 28 52.52 | −75 27 41.29 | 2.6 ± 0.3 | $1.07 \times 10^{-2}$ | 5.28 | 7.47 | 30.6 |
| Blob 2 † | 07 28 20.37 | −75 26 11.03 | - | - | - | - | - |
| Core † | 07 28 32.37 | −75 27 38.66 | - | - | - | - | - |
| **MKT J001940.4-654722** | | | | | | | |
| Blob 1 | 00 20 01.93 | −65 48 04.08 | 2.5 ± 0.1 | 3.84 | 1.54 | 4.04 | 48.1 |
| Blob 2 | 00 19 10.59 | −65 46 55.19 | 3.2 ± 0.4 | 2.69 | 22.99 | 15.60 | 51.5 |
| Core | 00 19 40.37 | −65 47 21.96 | 0.8 ± 0.1 | 2.04 | - | - | - |
| **ACO 548B** | | | | | | | |
| NE lobe | 05 45 22.10 | −25 47 29.88 | 2.4 ± 0.1 | 1.21 | 0.90 | 3.26 | 67.9 |
| SW lobe | 05 44 50.70 | −25 50 31.40 | 2.2 ± 0.1 | 1.93 | 0.91 | 3.11 | 66.9 |
| Core | 05 45 05.00 | −25 47 40.52 | 0.5 ± 0.1 | 1.03 | - | - | - |

We found that the magnetic field in our targets is of the order of a few μG. As mentioned by Rudnick [40], such low magnetic fields are not unique to dying radio galaxies. They are, however, exceptional in terms of what it is possible to observe. If the magnetic fields of dying radio galaxies drop below μG levels, the sources become very faint, and the lifetimes become very short due to inverse Compton losses.

Using a combination of synchrotron and inverse Compton cooling, we found that the radiative lifetime is of the order of 30–70 Myr for our sources. The age of the three observed dying radio galaxies is in agreement with the ages from modelling by Brienza et al. [5] and Godfrey, Morganti, & Brienza [41].

From a well-defined sample of 2215 AGN with $0.03 < z < 0.3$, Best et al. [42] found that the fraction of galaxies that host radio-loud AGN with $L_{1.4\text{GHz}} > 10^{23}$ W Hz$^{-1}$ is a strong function of stellar mass, rising from nearly zero below a stellar mass of $10^{10}$ $M_\odot$ to more than 30% at stellar masses of $5 \times 10^{11}$ $M_\odot$. Radio-loud AGNs are preferentially hosted by elliptical galaxies while radio-quiet ones are hosted by galaxies of later type [43]. The potential optical hosts of the three dying radio source presented in this study have masses between 1 and $100 \times 10^9$ $M_\odot$.

The optical counterparts of MKT J072851.2-752743 and ACO 548B show characteristics of low mass late type galaxies, while the proposed host for MKT J001940.4-654722 has a stellar mass more consistent thos of massive starburst galaxies.

## 5. Conclusions

Dying radio galaxies are expected to lack features typical of active galaxies, such as defined jets or hot spots, and depending on whether the central activity has ceased gradually or abruptly, they can also show weak or absent cores. We analysed three sources detected from the MGCLS images. The three objects analysed satisfy the criteria of dying radio galaxies.

Of the three sources, the most intriguing is ACO 548B, which had earlier been classified as two-relic system. We showed that this source is instead a dying radio galaxy due to the lack of hot ICM, which would show up in X-ray emission, the non-peripheral location near the centre of the cluster, non-elongated radio morphology, and having energetics characteristic of dying radio sources.

Dying radio galaxies are hard to find and require a telescope with both good sensitivity and resolution. We find that the MeerKAT is an ideal instrument for detecting such sources, especially in the lower-frequency MeerKAT bands. The higher-frequency bands can be used for better subtraction of contaminating unresolved sources in order to better study the diffuse emission, using techniques discussed by Scaife et al. [44]. Future targeted multi-wavelength observations of the potential dying radio sources will allow us to confirm their nature and characterise them fully.

**Author Contributions:** All the listed authors have contributed in this paper. Conceptualization, N.O.; methodology, N.O.; validation, N.O., L.R., M.F.B.; formal analysis, N.O.; investigation, N.O.; resources, N.O., L.R., M.F.B., T.V., K.K. (Kenda Knowles), K.K. (Konstantinos Kolokythas) and N.M.; data curation, N.O.; writing—original draft preparation, N.O.; writing—review and editing, N.O., L.R., M.F.B., T.V., K.K. (Kenda Knowles), K.K. (Konstantinos Kolokythas) and N.M.; visualization, N.O., L.R. All authors have read and agreed to the published version of the manuscript.

**Funding:** The MeerKAT telescope is operated by the South African Radio Astronomy Observatory (SARAO), which is a facility of the National Research Foundation, an agency of the Department of Science and Innovation. This research has been conducted using resources provided by the United Kingdom Science and Technology Facilities Council (UK STFC) through the Newton Fund and SARAO.

**Institutional Review Board Statement:** Not applicable.

**Informed Consent Statement:** Not applicable.

**Data Availability Statement:** The first MGCLS data release (DR1) will be made available at https://www.archive.sarao.ac.za (accessed on 1 June 2021). All legacy data products can be found at https://doi.org/10.48479/7epd-w356 (accessed on 1 June 2021).

**Acknowledgments:** The authors thank the anonymous referees whose comments have greatly improved the manuscript. The MGCLS data products were provided by SARAO and the MGCLS team. N.O will like to thank Matt Hilton for providing redshift data for some of the targets in this paper. T.V acknowledges the support from the Ministero degli Affari Esteri e della Cooperazione Internazionale, Direzione Generale per la Promozione del Sistema Paese, Progetto di Grande Rilevanza, ZA18GR02.

**Conflicts of Interest:** The authors declare no conflict of interest.

## Notes

1. We use $S \propto \nu^{-\alpha}$, where $S$ is the flux density, $\nu$ is the frequency, and $\alpha$ is the spectral index.
2. https://github.com/ACTCollaboration/zCluster (accessed on 1 June 2021).
3. http://www.cv.nrao.edu/~bcotton/Obit.html (accessed on 7 July 2021).

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
