# Peer review of "Discovery of Rare Dying Radio Galaxies Using MeerKAT"

_galaxies, doi:10.3390/galaxies9040102_

Round 1

Reviewer 1 Report

This paper presents interesting new data from the MeerKAT
telescope. The authors identify regions of diffuse radio emission in
fields surrounding clusters of galaxies from the survey being
undertaken by MeerKAT. The three sources identified are potentially in
the class of dying radio galaxies, giving the authors an opportunity
to explore the energetics, ages and powers of these three systems
after identifying the potential host galaxy for each.

Identification of new examples of dying radio galaxies is important
for understanding the evolution of AGN. The sensitivity and resolution
of MeerKAT are both well suited to this process as both the diffuse
and compact components can be identified with the same data. The
identification of the host galaxy for a potential dying radio galaxy
is complex though and while the authors clearly describe their
approach, the identifications are not without uncertainties. For this
reason I would recommend that the authors add the word 'potential'
within the abstract in two places: the first in the sentence "We
present the study of three _potential_ ..." and the second "The
_potential_ optical counterparts...". This softening of the abstract
would make it more consistent with the Discussion where the sources
are clearly described as 'potential dying radio sources'.

Given the great potential of MeerKAT data to identify and permit study
of dying radio galaxies, I suggest the authors consider adding a
wrapper statement at the end of the Abstract about what this could
mean for our understanding of the evolution of the AGN population.

The Introduction describes dying radio galaxies (AGN relics) and radio
relics but seems incomplete without some description of radio Phoenix
sources as originally described in Kempner et al. (2003) and more
recently discussed in the van Weeren et al. (2019) SSRv article.

There are a lot of assumptions made in identifying the host galaxy for
the diffuse radio sources presented here. Are the authors able to
estimate the probability of a chance identification of the host for
the system? This is not critical but could help strengthen the case
for the identification.

What are the prospect for MeerKAT to detect polarization for the
emission in ACO 548B to confirm the Feretti et al results? It seems
unlikely that polarization at this level be expected for a dying radio
source so this would be an important discriminator between relic and
dying radio source. 

I am not sure I understand what is being implied (if anything) by the
statement "if the diffuse radio lobes were part of a relic there would
be trace of some hot X-ray plasma within them". The X-ray connection
to relics can be missed if the X-rays are not deep enough but even
when the thermal emission covers the region of the relic there is
often not a clear hot X-ray component that is identified as tied to
the relic. This could be due to too few photons or a shock that is at
an angle to our line of sight.

Minor issues:

Figures: the axes in all figures are very difficult to read with small
text. Please consider re-making with much larger text.

Figure 1 -- missing '.' at end of 4'th sentence.

Figure 2 -- missing 'W' for WISE name.

Line 165: Section 3.1.2: Typo redhsift -> redshift

Line 205: should refer to Figure 3

Figure 5: The axes on this figure are particularly difficult to
read. The spectral index colorbar is not shown in the same notation as
the convention used for the paper. Please modify to the notation for
the paper.

Author Response

Reviewer 1:

This paper presents interesting new data from the MeerKAT

telescope. The authors identify regions of diffuse radio emission in

fields surrounding clusters of galaxies from the survey being

undertaken by MeerKAT. The three sources identified are potentially in

the class of dying radio galaxies, giving the authors an opportunity

to explore the energetics, ages and powers of these three systems

after identifying the potential host galaxy for each.

Identification of new examples of dying radio galaxies is important

for understanding the evolution of AGN. The sensitivity and resolution

of MeerKAT are both well suited to this process as both the diffuse

and compact components can be identified with the same data. The

identification of the host galaxy for a potential dying radio galaxy

is complex though and while the authors clearly describe their

approach, the identifications are not without uncertainties. For this

reason I would recommend that the authors add the word 'potential'

within the abstract in two places: the first in the sentence "We

present the study of three _potential_ ..." and the second "The

_potential_ optical counterparts...". This softening of the abstract

would make it more consistent with the Discussion where the sources

are clearly described as 'potential dying radio sources'.

Corrected

Given the great potential of MeerKAT data to identify and permit study

of dying radio galaxies, I suggest the authors consider adding a

wrapper statement at the end of the Abstract about what this could

mean for our understanding of the evolution of the AGN population.

Added

The Introduction describes dying radio galaxies (AGN relics) and radio

relics but seems incomplete without some description of radio Phoenix

sources as originally described in Kempner et al. (2003) and more

recently discussed in the van Weeren et al. (2019) SSRv article.

Added paragraph in bold 

There are a lot of assumptions made in identifying the host galaxy for

the diffuse radio sources presented here. Are the authors able to

estimate the probability of a chance identification of the host for

the system? This is not critical but could help strengthen the case

for the identification.

Added on Pg 14

What are the prospect for MeerKAT to detect polarization for the

emission in ACO 548B to confirm the Feretti et al results? It seems

unlikely that polarization at this level be expected for a dying radio

source so this would be an important discriminator between relic and

dying radio source. 

Comment: Section on polarization has been added in bold on Pg 11-12

I am not sure I understand what is being implied (if anything) by the

statement "if the diffuse radio lobes were part of a relic there would

be trace of some hot X-ray plasma within them". The X-ray connection

to relics can be missed if the X-rays are not deep enough but even

when the thermal emission covers the region of the relic there is

often not a clear hot X-ray component that is identified as tied to

the relic. This could be due to too few photons or a shock that is at

an angle to our line of sight.

Removed this sentence.

Minor issues:

Figures: the axes in all figures are very difficult to read with small

text. Please consider re-making with much larger text.

Figure 1 -- missing '.' at end of 4'th sentence.

Corrected

Figure 2 -- missing 'W' for WISE name.

Corrected

Line 165: Section 3.1.2: Typo redhsift -> redshift

Corrected

Line 205: should refer to Figure 3

Changed

Figure 5: The axes on this figure are particularly difficult to

read. The spectral index colorbar is not shown in the same notation as

the convention used for the paper. Please modify to the notation for

Changed the figure

Reviewer 2 Report

This manuscript presents the discovery of three dying radio galaxies, i.e., sources with fading radio lobes with no evidence of nuclear activity. Identifying and studying this type of radio sources is critical to get a more general understanding of the various stages of active galactic nuclei. The manuscript presents a robust physical characterization of the observed radio emission using available radio data. Likewise, all the information of the host galaxies that can be derived from existing data is reported. Yet, this reviewer has identified some minor issues regarding the presentation of the results.

  • Lines 33-34: The manuscript would benefit from a more detailed literature review on dying radio galaxies. I recommend elaborating, in particular, on the number of sources of this type that have been reported before. This is particularly important to support the title of the manuscript: “ Discovery of rare …”. For instance, I suggest mentioning the expected fraction and space density of dying radio galaxies that have been derived by Brienza+16 (arXiv:1603.01837v1) and Hurley-Walker+15 (https://doi.org/10.1093/mnras/stu2570).

  • Line 98: The manuscript should provide more information about how the search for non-cluster radio emission was performed, to address which criteria/parameters (in general) can be used to differentiate dying radio galaxies from diffuse cluster radio emission.  

  • Line 99: I recommend mentioning the number of images that were inspected to look for weak radio emission.

  • Line 114, 180, and 246: I suggest including the redshift of the three clusters.  

  • Line 122: I suggest rephrasing as “… revealed no radio counterpart”, to avoid confusion with the search for optical counterparts that is presented in the following section. 

  • Line 132: “they are outside the primary beam”. This reader finds this line to be misleading. I suggest rephrasing as: “they are outside the available primary-beam corrected image”. 

  • Line 142: Is this angular size given in terms of deconvolved FWHM? Is this radio size measured in the full-resolution radio image? I also suggest combining this information with that provided in line 152. 

  • Lines 171-172: I suggest providing a reference (or provide the equation) used to derive the magnetic field and total energy density, as it has been done for the radiative lifetimes. 

  • Lines 222-223: I recommend including the information about the library/software/package (and adopted input parameters) used to derive the photometric redshift. 

  • Line 274: Is the default alpha=-0.6 correct? This reader suspect that it should be +0.6 (following your adopted convention/definition of alpha). 

  • Fig. 5: The units of the colorbar appear to be in disagreement with the adopted convention/definition of alpha; specifically, this reader suspect that these should be positive values. 

  • Section 4: While the properties of the hosts are presented in the analysis/results sections, these are not further discussed in Section 4 (nor 5). The manuscript would benefit from a comparison of the hosts of “normal/active” and dying radio galaxies. 

MINOR SUGGESTIONS/RECOMMENDATIONS:

  • Figure 1, 2, 3, 4, 6: Tick labels, especially those of the colorbar, are too small. 

  • Fig. 1. “… beam). Two …”  

  • Line 88: “image centered at” 

  • Line 92: “ (RFI). These…”   

  • Some references appear to be missing in the introduction (first and second paragraph). For instance, in lines 23 and 24. Also, I suggest adding a reference to the statement about the mass of massive starburst in line 243.

  • Section 5: Do you have any potential follow-up study in mind (e.g., obtaining optical spectra of hosts or extending the search of dying radio sources)? If so, I would suggest mentioning this at the end of Section 5. 

  • The references do not follow the MDPI style. 

Author Response

Reviewer 2:

This manuscript presents the discovery of three dying radio galaxies, i.e., sources with fading radio lobes with no evidence of nuclear activity. Identifying and studying this type of radio sources is critical to get a more general understanding of the various stages of active galactic nuclei. The manuscript presents a robust physical characterization of the observed radio emission using available radio data. Likewise, all the information of the host galaxies that can be derived from existing data is reported. Yet, this reviewer has identified some minor issues regarding the presentation of the results.

Lines 33-34: The manuscript would benefit from a more detailed literature review on dying radio galaxies. I recommend elaborating, in particular, on the number of sources of this type that have been reported before. This is particularly important to support the title of the manuscript: “ Discovery of rare …”. For instance, I suggest mentioning the expected fraction and space density of dying radio galaxies that have been derived by Brienza+16 (arXiv:1603.01837v1) and Hurley-Walker+15 (https://doi.org/10.1093/mnras/stu2570).

 Added in bold Pg 2 ln 36-42

Line 98: The manuscript should provide more information about how the search for non-cluster radio emission was performed, to address which criteria/parameters (in general) can be used to differentiate dying radio galaxies from diffuse cluster radio emission.  

Added in bold Pg 3 ln 121 onwards

Line 99: I recommend mentioning the number of images that were inspected to look for weak radio emission.

Added the number (115)

Line 114, 180, and 246: I suggest including the redshift of the three clusters.  

Added as per reviewer’s comments

Line 122: I suggest rephrasing as “… revealed no radio counterpart”, to avoid confusion with the search for optical counterparts that is presented in the following section. 

Corrected

Line 132: “they are outside the primary beam”. This reader finds this line to be misleading. I suggest rephrasing as: “they are outside the available primary-beam corrected image”. 

Corrected

Line 142: Is this angular size given in terms of deconvolved FWHM? Is this radio size measured in the full-resolution radio image? I also suggest combining this information with that provided in line 152. 

Corrected

Lines 171-172: I suggest providing a reference (or provide the equation) used to derive the magnetic field and total energy density, as it has been done for the radiative lifetimes. 

Added references

Lines 222-223: I recommend including the information about the library/software/package (and adopted input parameters) used to derive the photometric redshift. 

Added 

Line 274: Is the default alpha=-0.6 correct? This reader suspect that it should be +0.6 (following your adopted convention/definition of alpha). 

Corrected 

Fig. 5: The units of the colorbar appear to be in disagreement with the adopted convention/definition of alpha; specifically, this reader suspect that these should be positive values. 

Figure 5 has been re-made to address the reviewer's comment.

Section 4: While the properties of the hosts are presented in the analysis/results sections, these are not further discussed in Section 4 (nor 5). The manuscript would benefit from a comparison of the hosts of “normal/active” and dying radio galaxies. 

Added in bold on Pg 14  lines 379-488

MINOR SUGGESTIONS/RECOMMENDATIONS:

Figure 1, 2, 3, 4, 6: Tick labels, especially those of the colorbar, are too small. 

Font sizes have been increased

Fig. 1. “… beam). Two …”  

Corrected

Line 88: “image centered at” 

Corrected

Line 92: “ (RFI). These…”   

Corrected

Some references appear to be missing in the introduction (first and second paragraph). For instance, in lines 23 and 24. Also, I suggest adding a reference to the statement about the mass of massive starburst in line 243.

Added in bold 

Section 5: Do you have any potential follow-up study in mind (e.g., obtaining optical spectra of hosts or extending the search of dying radio sources)? If so, I would suggest mentioning this at the end of Section 5. 

Added correction line 405-406

The references do not follow the MDPI style. 

All references have been corrected to reflect the MDPI style.